# Grappling with Gulf War Illness: Perspectives of Gulf War Providers

**DOI:** 10.3390/ijerph17228574

**Published:** 2020-11-19

**Authors:** Girija Kaimal, Rebekka Dieterich-Hartwell

**Affiliations:** Creative Arts Therapy Department, Drexel University, Philadelphia, PA 19102, USA; gk27@drexel.edu

**Keywords:** Gulf War illness, health care providers, qualitative research, grounded theory method

## Abstract

*Background*: Although the Gulf War occurred almost 30 years ago, the chronic symptoms of Gulf War illness (GWI), which include respiratory, gastrointestinal, and skin problems, as well as fatigue, pain, and mood alterations, currently affect over 200,000 veterans. Meanwhile, healthcare providers lack clear guidelines about how to best treat this illness. The objective in this study was to learn about the perceptions and experiences of healthcare providers of GWI veterans in terms of medical symptoms, resources for treatment, and quality of care. *Methods*: We interviewed 10 healthcare providers across the United States and subsequently conducted a qualitative grounded theory study which entailed both systematic data analysis and generating a grounded theory framework. *Results*: Our findings indicated multiple challenges for providers of veterans with GWI, including gaps in knowledge about GWI, lack of treatment options, absence of consistent communication within the Department of Veterans Affairs (VA) system, and personalized care that was limited to validation. *Conclusion*: While this study had several limitations, it supported the notion that healthcare providers have inadequate knowledge and awareness about GWI, which leads to continued uncertainty about how to best care for GWI veterans. This could be remedied by the creation of a comprehensive curriculum for a Massive Open Online Course (MOOC) to serve as an educational tool for those attending to this largely overlooked veteran population.

## 1. Introduction

Between August of 1990 and February of 1991, the United States (US) and over 30 coalition countries deployed troops to the Persian Gulf under Operation Desert Storm and Operation Desert Shield, a mission launched in opposition to Iraq’s invasion of Kuwait. In total, nearly one million service members, among them 700,000 troops from the US, served in these short-term yet large-scale efforts. Many were exposed to etiological agents, such as fumes of oil well fires, pesticides including carbamates and organophosphates, and other toxins [1,2]. Furthermore, the muscle-strengthening drug pyridostigmine bromide was routinely administered as a nerve agent prophylaxis [3]. Within months of their return, many service members began to report multiple symptoms that were difficult to explain. Symptoms included widespread pain, muscle aches, headache, fatigue, respiratory problems, gastrointestinal issues, memory and cognitive defects, and skin abnormalities, as well as mood alterations [4]. In addition, service members frequently manifested changes in behavior, as well as problems with interpersonal relationships [5].

Throughout the following decades, no clear etiology or medical recognition of Gulf War illness (GWI) was established, and many Gulf War veterans felt invalidated and frustrated [6]. A report by the Government Accountability Office showed that the Department of Veterans Affairs (VA) approved only 17% of claims for compensation for veterans with GWI between 2011 and 2015, three times lower than all other claimed disabilities during this time [7]. Furthermore, Gulf War veterans seeking benefits had to wait four months longer on average to hear back from the VA at that point [7].

In 2014, the National Academy of Medicine advocated for researchers and clinicians to use a particular set of criteria to identify Gulf War veterans with chronic multisymptom illness [8]. This advocacy and the associated book outlined GWI as a specific illness that falls under the chronic multisymptom illness (CMI) umbrella and has two official case definitions, one put forth by investigators for the Centers for Disease Control and Prevention (CDC) and the other being the Kansas criteria. Between 29% and 60% of Gulf War veterans meet the CDC’s criteria for GWI, while 34% of veterans meet the Kansas criteria (Kansas criteria: three of six domains: (1) fatigue and sleep problems, (2) pain symptoms, (3) neurologic, cognitive, and/or mood symptoms, (4) gastrointestinal symptoms, (5) respiratory symptoms, and (6) skin symptoms [8]). As per the more inclusive and general CDC definition (CDC criteria: one or more from at least two of the following categories: (1) fatigue, (2) mood and cognition (symptoms of feeling depressed, difficulty in remembering or concentrating, feeling moody, feeling anxious, trouble in finding words, or difficulty in sleeping), and (3) musculoskeletal (symptoms of joint pain, joint stiffness, or muscle pain) [9]), veterans are diagnosed with GWI if they report one or more symptom(s) that last for 6 months or longer in two of three categories: fatigue, musculoskeletal pain (joint pain, joint stiffness, or muscle pain) and mood/cognition (depression, difficulty in remembering, anxiety, difficulty in sleeping, etc.) [9]. According to a recent meta-analysis of self-reported health symptoms in Gulf War veterans, the most commonly reported symptoms were fatigue, pain, cognitive and mood problems, skin rash, gastrointestinal issues, and respiratory concerns [1]. This symptom complex lined up with the more specific Kansas definition which identifies GWI in those who report multiple and/or moderate levels of symptoms in three of six categories in the year before the assessment: fatigue/sleep, pain, neurological/cognitive/mood, respiratory, gastrointestinal, and skin [10]. A third more restrictive definition, put forth by Haley (Haley criteria: five of eight signs or symptoms: (1) fatigue, (2) arthralgia or lower back pain, (3) headache, (4) intermittent diarrhea without bloody stools, (5) neuropsychiatric complaints of forgetfulness, difficulty in concentrating, depression, memory loss, or easy irritability, (6) difficulty in sleeping, (7) low-grade fever, and (8) weight loss [11]), includes three symptom complexes: syndrome 1 (compromised cognition) entails problems with attention, memory, sleep, and depression; syndrome 2 (confusion/ataxia) is characterized by thinking and balance symptoms; syndrome 3 (neuropathic pain) requires self-reported joint and muscle pain [11]. While the CDC and Kansas definitions are officially recognized and most commonly used by researchers and clinicians, the Haley syndrome criteria are not accepted by the National Academy of Medicine.

Given varying locations of deployment and different countries of units with subsequent varying exposures, no single cause of GWI exists; instead, a number of factors are believed to play a role in the pathogenesis of GWI [12,13,14]. For example, the heavy use of pesticides intended to prevent arthropod-borne infectious diseases resulted in overexposure to 15 potentially toxic substances of approximately 40,000 US service members, including organophosphates, carbonates, pyrethroids, and highly concentrated DEET (*N*,*N*-diethyl-*meta*-toluamide), an oil common in insect repellant [15]. Prophylactic pyridostigmine bromide (PB) pills were distributed to the US, United Kingdom (UK), and Canadian troops as a nerve agent pretreatment against possible exposures during an attack by enemy troops [16]. These PB pills, believed to have been distributed to up to 400,000 military personnel, have been found to inactivate important enzymes and cause altered gene expression and delayed cognitive symptoms [12]. Service members were also exposed to dioxins and furans in smoke from over 65 oil well fires, and about 250,000 US soldiers were exposed to chemicals, such as sarin, which can cause eye, skin, and respiratory damage, as well as a range of systemic effects [12].

Today, over 200,000 deployed veterans, approximately 30%, continue to be affected by the chronic symptoms of GWI [1]. It is clear from the literature that there is causal connection of exposures to GWI [17,18]. Furthermore, organic pathologies and mechanisms have been well documented and replicated, including inflammation, coagulation activation, mitochondrial impairment, white matter injury, and immune dysregulation among others [19]. However, the lack of clarity and widespread knowledge about symptoms and exposures complicate both diagnosis and treatment of GWI and leave healthcare providers without clear guidelines. Most frequently, GWI is treated as other chronic multisymptom illnesses, such as fibromyalgia, chronic fatigue syndrome, and functional gastrointestinal disorders [20]. Some clinical but preliminary practical guidelines include cognitive behavioral therapy, acupuncture, and the use of coenzyme Q10 [21,22,23] for alleviation of some of the symptoms faced by veterans as a result of GWI. The use of antidepressants, while recommended by the VA [21], has not shown any clear benefits. To this date, no standard level of care exists [2,5,22,23,24,25]. Healthcare providers of GWI veterans are without clear treatment direction. Furthermore, there are no published studies highlighting the perspectives of healthcare providers serving this population.

Our objective in this study was to learn about the perceptions and experiences of healthcare providers who have cared for GWI veterans in terms of medical symptoms, resources for treatment, and quality of care. We wanted to understand healthcare providers’ perceptions, as well as what support and resources healthcare providers need to better serve this population.

## 2. Materials and Methods

### 2.1. Study Population

Participants were recruited through snowball sampling, a method wherein participants already interviewed refer additional potential interviewees. Initial referrals came from collaborators working in the VA system. Fifteen potential interviewees were identified; 10 responded to the request for participation and five did not reply to requests to schedule a time. We interviewed 10 healthcare providers of veterans with GWI from across the United States via an HIPAA (Health Insurance Portability and Accountability Act) -protected Zoom video platform. Participants included providers from a range of settings including hospitals and health centers. Most of the participants had direct patient contact with up to 30 years of experience, while others were in a supervisory or research role. In all, interview respondents encompassed three men and seven women in the following positions: physician (*n* = 3), psychologist (*n* = 2), nurse (*n* = 1), acupuncturist (*n* = 2), social worker (*n* = 1), medical sociologist (*n* = 1), and pharmacologist (*n* = 1). One person identified as both nurse and acupuncturist. Their identified race/ethnicity was White (*n* = 8) and Asian (*n* = 2). All participants provided informed consent. The protocol was approved by the institutional review board at Drexel University (IRB ID: 1804006244), Philadelphia, as well as the Human Resource Protection Office of the US Army Medical Research and Development Command.

### 2.2. Setting

The interviews took place via Zoom as participants were located in different parts of the United States. We utilized a semi-structured interview guide, specifically developed for this study on the basis of a comprehensive literature review, and interviews lasted between 30 and 45 min. We asked about the perceptions and experiences of healthcare providers who had cared for (or were currently caring for) veterans with GWI in terms of medical symptoms, resources for treatment, and quality of care. Sample questions were the following: “What would you say are the clinical features of Gulf War Illness?”, “What kind of medical care do veterans with Gulf War Illness receive?” “What kinds of resources would you like to see offered?”, etc. The semi-structured interview protocol can be found in Appendix A.

### 2.3. Data Collection

The interviews were conducted using Zoom which is a HIPAA secure digital teleconferencing tool. The interviews were all conducted and recorded by the first author. Participants completed informed consent and were offered $25 for participation. However, only those participants who were not employed by the VA were eligible to accept the compensation while participants who were VA employees declined it. The recorded interviews were transcribed by the second author and cross-checked for accuracy in transcription by two additional members of the research team. The transcriptions were entered into Dedoose, a software platform used to analyze qualitative and mixed methods research data.

### 2.4. Data Analysis

We utilized a qualitative research method, more specifically, a grounded theory method [26], to analyze the interview data. This type of analysis involves the creation of an emergent theory or framework for a problem or phenomenon through systematic, iterative analysis of data [26]. A preliminary coding scheme based on the research question “What are the perceptions and experiences of health care providers who serve veterans with GWI in terms of medical symptoms, resources for treatment, and quality of care?” was developed, refined over the course of several transcripts, and reviewed with one transcript by two coders to ensure shared understanding of the codes. An intercoder agreement of 90% was obtained over the course of the data analysis. The data were coded in several phases following the method of Corbin and Strauss [26]. These phases were open coding, an initial stage that entailed line-by-line coding of the transcripts and developing categories and subcategories for corresponding codes, axial coding, a stage during which the connections between categories and their sub-categories were established, and selective coding, a stage that entailed revisiting the transcripts to validate the main categories which were identified based on recurring patterns in the data. Throughout data analysis, a process of constant comparison of codes and categories, as well as memo-writing, provided an illustration for emerging ideas and enabled the creation of the theoretical framework.

## 3. Results

The findings indicated that healthcare providers were challenged by four aspects related to their roles when serving veterans with GWI. These included (1) gaps in knowledge of the multisymptom illness, (2) limited treatment options, (3) challenges in personalizing care, and (4) systemic barriers to providing health care GWI veterans. The figure below summarizes the key categories in a theoretical framework. As seen in Figure 1, healthcare provider participants felt challenged in all aspects of their role in serving patients with GWI.

### 3.1. Gaps in Knowledge of the Multisymptom Illness

Overall, there were gaps in the understanding of Gulf War Illness, and even those with expertise on the topic recognized the challenges in defining the features of this multisymptom illness. All the participants identified challenges in knowledge of both the toxic exposures and their effects on GW veterans.


*“There is a kind of general sense that it might have something to do with toxic exposures, that many of these soldiers, many of these active duty service members were exposed to burning oil wells and burning pits and they may have inhaled something toxic.” (White, male, physician)*



*“It’s hard to look at, you know… a syndrome when no one is like, you know saying, this is the cause. You know? You know it’s still even hard now to talk about what it is, and what is it going to turn into.” (White, female, psychologist)*



*“We have no way of knowing, ah, whether there’s a causal relationship with these exposures.” (White, male, physician)*



*“Most people don’t know for sure [about the exposures]. And a small number of people when they say yes, you know, they really are not aware of what they are saying.” (Asian, female, physician)*



*“Most healthcare providers do not know. It’s less understood than it was. The information that we know is not getting down to the clinicians.” (White, female, pharmacologist)*



*“Sometimes it’s difficult to sift out what are the symptoms, like mood symptoms are related to trauma or they are related to toxic exposure, or does the toxic exposure put somebody at higher risk to be susceptible to trauma, you know, it’s really hard to disentangle and understand all these things.” (White, female, psychologist)*


### 3.2. Limited Treatment Options

The participants further identified a lack of established viable treatment options that cover all aspects of the illness and a tendency to treat symptoms alone treatment options and a tendency to treat symptoms alone.
“There are no Food and Drug Administration (FDA)-approved treatments for Gulf War illness. So even if they were identified, right now you’re just kind of piecemeal treating the symptoms. So maybe you’re treating the mood symptoms, maybe you’re treating the chronic pain, maybe you’re treating the skin rashes, so that’s [a] barrier to care that we don’t have any FDA-approved treatments.” (White, female, psychologist)
“I think that they are being treated… not holistically, but they’ll go in and say, hey, you know I’ve got stomach problems, and they’re just addressing that, they’re not looking at their overall whole-body condition.” (White, female, social worker)
“We kind of approach things symptomatically. As far as I know that’s kind of the approach people take. You know, if they have pain, you try to give medications that are good for chronic pain. If they have problems concentrating, you can give them kind of medication that boosts their concentration.” (White, male, physician)

The acupuncturists interviewed spoke about a more holistic approach, but they also identified challenges and limitations in their practice.
“When I was doing my acupuncture points, I treated the whole organ. Not just anti-inflammatory, and analgesic, not just that. I treated the whole organ.” (Asian, female, nurse and acupuncturist)
“Right now, it’s a confusing process for the acupuncturists to get compensated and a lot of the VAs don’t even tell people that they can be compensated, so it’s really a payment thing.” (White, female, sociologist)

### 3.3. Challenges in Personalizing Care

One of the particular challenges addressed by the participants was the lack of personalized care options. Providers generally seemed to not know enough to address aging, genetic issues, and differences by gender.
“There’s going to be genetic differences in susceptibility. So, some people may be more susceptible to the toxins they were exposed to, to present certain symptoms, as opposed to others.” (White, female, pharmacologist)
“Yea, I think women in the military who have GWI is more complicated because of the hormone balance. Hormones have a big thing to do with a lot of their symptoms. It aggravates them. You know men have this androgen and estrogen issues thing too, but I think it’s easier for men.” (Asian, female, nurse and acupuncturist)
“We are running into a lot of osteoarthritis. That’s probably the main thing in the pain study. A lot of people are running into a lot of cognitive problems and it’s hard to say if its due to aging or if you know, it’s accelerated the aging process, um, so yeah.” (White, male, physician)

Several participants identified the importance of validation of the patient experience even in the midst of the lack of established treatment options.
“No one’s been there to listen to them about all these problems that they’re facing, you know they kind of go to a separate doctor for each ailment that they have and that’s really problematic.” (White, female, psychologist)
“Some kind of standard way of validating you know, ‘even if there is nothing we can do for you, we acknowledge that this happened. And, um, that you are struggling with symptoms.’ I think that would go a long way. I think the impulse is to say, well I don’t know what to do so let’s not talk about it. Um, and so… some kind of psychological approach to what is going on may be helpful in the absence of being able to kind of attack the root causes of the biological problems.” (White, male, physician)

### 3.4. Systemic Issues

Underlying the challenges was a systemic issue, beginning with a lack of general recognition of GWI.


*“I’m not sure that I would say that it’s real, because I don’t think we really know what it is. And I think there are some people who are pretty skeptical of it… no specific pathology has been proven.” (White, male, physician)*



*“I think because of the early beginnings of skepticism, an awful lot of VA providers still don’t believe it or don’t have the education to know that it’s real or even know what it is.” (White, female, pharmacologist)*



*“I just feel like there’s, you know, it’s been so limited in the ability to, you know, to have people, even be willing to open up to the idea that this exists. And so, for me anyway, we’ve, you know, I’ve been to meetings where people have basically said to me, ‘oh you’re lying, ok move on’.” (White, female, psychologist)*


One participant mentioned that there was no place in the medical records to enter GWI as an illness.


*“So, in the VA we use a computerized patient record system or CPRS and it lists the problems that the patient has, it lists the diagnoses. I can’t recall seeing on a list of diagnoses Gulf War illness.” (White, male, physician)*


Furthermore, one participant reported that there were no specific benefits to being recognized as GWI veteran besides the registry:


*“So, and it’s not clear to me what the benefits for the patient are to have a Gulf War designation, though maybe it changes their service connection which could be helpful to a patient, but I don’t actually know that… it doesn’t do anything for the patient in terms of entitlement and it doesn’t do anything for the patient in terms of treatment.” (White, male, physician)*


Overall, participants identified a disrupted and inconsistent communication from the top down.


*“The information we know is not getting down to the clinicians. There’s some lack of communication between what’s happening in congress and what filters down to the VA and what filters down to the healthcare system versus the benefits system, because benefits are separate than the healthcare system. There’s a roadblock.” (White, female, pharmacologist)*



*“We have research, but it’s just getting it out to people… There is also a lack of consistency across VAs, that everybody talks about. The best analogy I’ve heard is that they’re like little fiefdoms, which they are; they have different policies and different things going on within each of them… and so some more shared systems across VAs would make it easier to have quality programs.” (White, female, sociologist)*


Several participants spoke of GWI veterans as a group on the fringes of the system.


*“The overwhelming thing is they have been ignored by the system.” (White, male, physician)*



*“[Gulf War veterans] are kind of like the invisible veterans.” (White, female, social worker)*



*“I think that… the biggest problem early on was that no one believed them. And, that it was harder and harder to find people that looked at it as a global disorder, right?” (White, female, pharmacologist)*



*“I know, these vets had a lot of trouble even getting recognized as a syndrome and then, the history of the military is a history of not wanting to, once something is labelled you have to do something about it.” (White, female, sociologist)*


When asked about what supports and resources healthcare providers perceive they need to better serve this population, participants identified a need for more information, resources, and guidance on how to address the illness.


*“I’d like to know more about treatment. What has been considered successful. I know they are doing a study up there in Waco to see if maybe there’s certain medications that might help some of the symptoms. So, I would like to know more about medical treatment for it.” (White, female, social worker)*



*“So, from a curiosity standpoint and from future tracking and studies, it would be helpful I think if they came in with DEET exposure or uranium exposure or oil field exposure or whatever else that would be helpful for tracking and for my own curiosity, but not necessary, not necessary for a good acupuncture physician.” (White, male, acupuncturist)*



*“It would be nice if the ones who have to deal with the veteran care, if they know a little bit more. It would be, I mean not little bit, as much as they can.” (Asian, female, physician)*



*“I think that a lot the education for providers needs to be about you know kind of slowing down and trying to figure out if this could be more of an overarching diagnosis that’s missing in their clinical picture. And then, you know, how to explain that to them like that they’ve been living with this for however many years and it’s just recognized now.” (Asian, female, psychologist)*



*“Oh definitely online… I guess some sort of forums, yeah, or training protocols, education forums to share more further ideas, case study you know stuff like that. (White, male, acupuncturist)*


## 4. Discussion

The results of this qualitative grounded theory study indicated that healthcare providers perceive and experience multiple challenges as they strive to serve veterans with GWI. These include gaps in knowledge of the illness, limited treatment options, challenges in personalizing care, and systemic issues. The findings also highlighted the need for better resources and support to be made available to health care providers in order to better serve veterans with GWI.

Our results confirmed the historical challenges and gaps encountered in the literature with regard to the course of illness progression in GWI condition, which has resulted in Gulf War veterans feeling invalidated and frustrated [6]. According to the interviews, healthcare providers lacked a fundamental understanding about the nature and causes of GWI. For example, providers were unsure about the relationship between toxic exposures and symptoms. They also were struggling to understand the disparity in encountered symptoms, especially as patients were also dealing with aspects of aging and illness progression over a long period of time.

Gaps in knowledge about the illness in turn appeared to be directly linked to a range of treatment options and approaches, substantiating the sentiment in the literature about the lack of a standard level of care [22,24]. Among the 10 healthcare providers, some opted for addressing and treating one symptom at a time (usually the reason for referral), whereas others employed more holistic approaches that regarded the body as a whole, indicating variations in treatment approaches.

The findings regarding challenges in personalizing care underscored the need to understand more about differences in gender, age, ethnicity, and rank. This is particularly important considering that ground forces were more exposed to unique environmental conditions than any other service members during the Gulf War. Ground forces in turn are made up of mostly enlisted ranks who generally have lower levels of education and less access to resources [27]. According to the interviews, validation of clients’ experiences appeared to be an integral part of treatment, particularly in light of limited treatment options. This finding substantiated Greenberg et al.’s [6] conception that it is crucial for healthcare providers to understand, recognize, and validate veterans’ unique experiences with GWI in order to build trust.

Systemic issues were found to be a big part of the challenge when caring for GW veterans. Veterans with GWI have a long history of neglect and accusations of malingering. Furthermore, a diagnostic code and benefits are still not recognized in the VA. Missing clear and consistent communication from the top down can lead to discrepancies in knowledge and information. This is especially relevant considering that non-VA providers may also serve veterans with GWI. Incongruities in knowledge and connection, thus, underscore the importance of transparent communication and consistency and call for more widespread dissemination of information.

According to the interview responses, there are gaps in knowledge and understanding about GWI among healthcare providers. Our short-term recommendations are, therefore, to provide readily accessible educational materials for healthcare providers, including presentations and papers. A long-term recommendation would be to create a comprehensive curriculum for a Massive Open Online Course (MOOC), incorporating the literature and supported by the most up-to-date evidence, to serve as a research-informed and up-to-date educational tool for healthcare providers attending to this overlooked veteran population. This curriculum would be made available to all healthcare providers treating GWI veterans in and outside of the VA. In order to create the curriculum, an in-depth study of both provider and veteran responses, as well as an extensive literature review, would be necessary. Some additional recommendations for specific areas for further study include examining differences in GWI presentation among gender, ethnicity, and rank as mentioned previously. A repetition of this study on a larger scale with more direct care provider (specifically nurses) interviews would also be beneficial in elucidating our findings. Lastly, a qualitative study with veterans with GWI would be important in learning more about their experiences, foregrounding their voices, and understanding their needs and concerns regarding quality of life and quality of care received.

Overall, an increased understanding about toxicity and the results of toxic exposures experienced during the GW can be applied to the GW population, as well as lead to an increased understanding about more recent combat, such as OEF (Operation Enduring Freedom) and OIF (Operation Iraqi Freedom) who have also been exposed to toxic substances.

One of the strengths of this study was that it was one of the first to document challenges faced by healthcare providers serving GWI veterans and to learn about their needs in order to improve healthcare for this population. However, there were several limitations. This was a small-scale, preliminary qualitative study with only 10 participants. Furthermore, although we attempted to recruit a diverse sample of providers (race/ethnicity, profession, gender) in order to represent a range of perspectives, we were only able to interview a few with direct patient contact and, in particular, we were limited to only one nurse participant. Another weakness of the study was that there was only a single interviewer. Thus, this represents preliminary impression, and the findings are not generalizable.

## 5. Conclusions

This study supported the notion that there are gaps in knowledge about GWI among healthcare providers, which leads to continued uncertainty about how to best treat GWI veterans. Given the challenges in care faced by veterans with GWI, further research including on systemic issues with lack of communication from the top down, and on difficulties in personalizing care is needed. The findings indicate that it is imperative that providers have access to educational materials on GWI. We recommend providing readily accessible educational resources including presentations and articles to healthcare providers. A long-term recommendation is to create an evidence-based and current curriculum for an MOOC to serve as an educational tool for healthcare providers in and outside of the VA system attending to this overlooked and underserved veteran population.

## Figures and Tables

**Figure 1 ijerph-17-08574-f001:**
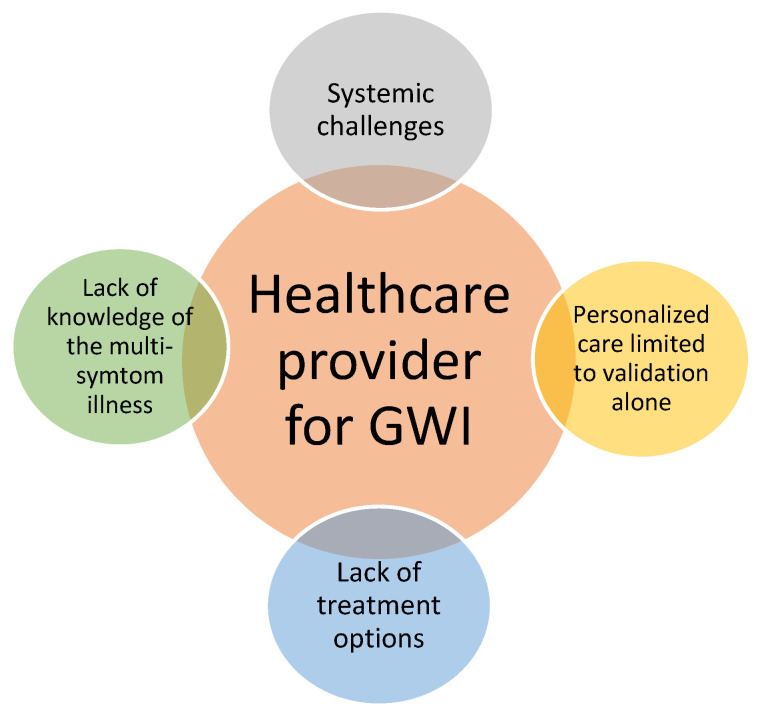
Challenges for healthcare providers serving GWI veterans.

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
