# Peer review of "Grappling with Gulf War Illness: Perspectives of Gulf War Providers"

_ijerph, 2020, doi:10.3390/ijerph17228574_

Round 1
Reviewer 1 Report
Line 13:
Explain Gulf War Illnesses (GWI) before you state the objective. Your literature review is very shallow.
Line 18:
Define VA Line 52&53 Give footnote for CDC criteria. Give footnote for Kansa criteria Give footnote on Haley.
Line:74 DEET not defined. Line122: Payment of $25 for participants is it ethically correct/ acceptable? Please verify.
What is the relevance of the race or color of the respondents? In Research Ethics it is not acceptable. This is study needs Ethical Approval from a recognized academic institution before being published.
As you rightly mentioned, the study is preliminary. How do you see this with your recommendation for a unified curriculum for a Massive Open Online Course 335 (MOOC)? Curriculum design recommendation requires an in-depth study.
The conclusion is very brief. Rather than recommending a new curriculum design ( maybe a long term solution), it will be more appealing to suggest alternative short term solutions for the problem under discussion.

Author Response
Point 1: Explain Gulf War Illnesses (GWI) before you state the objective. Your literature review is very shallow.
Response 1: We have included an expanded description of Gulf War Illness in the Literature Review section of the abstract (Line 10,11).
Point 2: Define VA
Response 2: We have defined VA as a short form for Department of Veterans Affairs (Line 19-20, Line 45).
Point 3: Give footnote for CDC criteria. Give footnote for Kansas criteria Give footnote on Haley.
Response 3: We have added footnotes for each criterium, including Haley (bottom of p.2).
Point 4 : DEET not defined.
Response: We have defined DEET as N,N-Diethyl-meta-toluamide (Line 77-78).
Point 5: Payment of $25 for participants is it ethically correct/ acceptable? Please verify.
Response 5: The $25 were offered to all the participants. However, only those participants who were not employees of the VA were eligible to accept the $25 compensation (3 participants). Participants who were VA employees declined the compensation. This has been clarified in the text (Line 129-131).
Point 6: What is the relevance of the race or color of the respondents? In Research Ethics it is not acceptable. This study needs Ethical Approval from a recognized academic institution before being published.
Response 6: This study received ethics approval from both Drexel University and HRPO which we have clarified in the text (Line 113-115). The relevance of race of the participants was related to the aims of the grant funded study at large, which was to explore differences related to race, gender, or ethnicity. The study did not discriminate against any participants, it merely noted the backgrounds of participants as well as veterans whom they served.
Point 7: As you rightly mentioned, the study is preliminary. How do you see this with your recommendation for a unified curriculum for a Massive Open Online Course 335 (MOOC)? Curriculum design recommendation requires an in-depth study.
Response 7: We are recommending a MOOC because, based on the interview responses, there is a lack of knowledge and understanding about GWI in health care providers. In order to create the MOOC curriculum, we are also collecting responses from 40 veterans with GWI in order to make this more comprehensive and representative. We have added this as long-term solution in the text (Line 360, Line 364-365).
Point 8: The conclusion is very brief. Rather than recommending a new curriculum design (maybe a long-term solution), it will be more appealing to suggest alternative short-term solutions for the problem under discussion.
Response 8: Thank you for this comment. We have added a short-term solution, i.e. readily accessible educational materials in the conclusion section (see Line 388-389).

Reviewer 2 Report
This is an interesting article, in which the authors intend to learn about the perceptions and experiences of health care providers of GWI veterans in terms of medical symptoms, resources for treatment, and quality of care.
Comments and suggestions:
- Abstract:
- What is the meaning of the abbreviation VA?
- The last sentences is not in line with the objetive of the study
- Materials and methods:
- In general, this section should be reviewed and explained in greater depth.
- Could paying the people who participated in the study affect the results obtained? How did the authors reduce this possible bias? It can be considered as a possible limitation of the study
- Was the semi-structured interview protocol based on any previous study?
- which was the computer software used for the analysis of the results?
- Discussion:
- As the materials and methods section, the discussion should be reviewed and axplained in greater depth.
- This section should be in line with the obtained results
- What is the meaning of the abbreviation OBE or OIF?
- How the authors believe that age, sex, ethnicity or professional type can influence the results obtained. This aspect should be compared with other studies.
- The specific limitations and strengths of this study should be indicated
- Discussion:
- The last sentences is not in line with the objetive of the study. It should be eliminated
Author Response
Point 1: What is the meaning of the abbreviation VA?
Response 1: We have added the meaning of the abbreviation (Department of Veterans Affairs) (Line 19-20, Line 45).
Point 2: Could paying the people who participated in the study affect the results obtained? How did the authors reduce this possible bias? It can be considered as a possible limitation of the study.
Response 2: We used standard practices for participant compensation. Important to note is that a majority of the participants declined compensation due to their employment status within the VA. Only those who were not affiliated with the VA accepted the $25. We have added this in the text (Line 129-131).
Point 3: Was the semi-structured interview protocol based on any previous study?
Response 3: This interview was developed for this study based on a comprehensive literature review. We have added this information to the manuscript (Line 118-119).
Point 4: which was the computer software used for the analysis of the results?
Response 4: we used the software program Dedoose, this is mentioned in the text (Line 133).
Point 5: As the materials and methods section, the discussion should be reviewed and explained in greater depth. This section should be in line with the obtained results
Response 5: We have sequenced the discussion section to be in line with the obtained results (Limited knowledge of the illness (Line 331-334), limited treatment options (Line 335-339), challenges in personalizing care (Line 340-348), systematic issues (Line 349-356).
Point 6: What is the meaning of the abbreviation OBE or OIF?
Response 6: OEF stands for Operation Enduring Freedom (war in Afghanistan) and OIF stands for Operation Iraqi Freedom (war in Iraq). We have added this information to the manuscript (Line 374-375).
Point 7: How the authors believe that age, sex, ethnicity or professional type can influence the results obtained. This aspect should be compared with other studies.
Response 7: We tried to get a diverse sample of respondents from different regions int he US in order to represent a range of perspectives, not because we thought this would influence the results (see Line 380-381). We are not aware of any studies where backgrounds of healthcare providers serving GWI veterans are available.
Point 8: The specific limitations and strengths of this study should be indicated
Response 8: We have added this in the discussion section (Line 376-383): Strengths: one of the first studies to document the challenges faced by health care providers serving GWI and how health care can be improved by identifying the needs of health care providers. Main Limitation: A larger sample of providers with direct patient contact.
Point 9: The last sentences is not in line with the objective of the study. It should be eliminated
Response 9: Thank you for this comment. We have retained the last sentence since it is a recommendation based on the findings of the study. We have also added a short-term recommendation to make it more appealing for the problem under discussion (Line 388-389). Furthermore, one of the long-term goals of the grant is to develop a MOOC so it needs to remain in the text.
